# A Case Report of Early-Onset Alzheimer’s Disease Using ^18^F-FDG PET and ^18^F-FBB PET

**DOI:** 10.3390/diagnostics13101671

**Published:** 2023-05-09

**Authors:** Jang Yoo, Miju Cheon, Min-Ju Kang

**Affiliations:** 1Department of Nuclear Medicine, VHS Medical Center, Seoul 05368, Republic of Korea; 2Department of Neurology, VHS Medical Center, Seoul 05368, Republic of Korea

**Keywords:** early-onset Alzheimer’s disease, ^18^F-FDG PET, ^18^F-florbetaben PET, case report

## Abstract

We describe a 40-year-old female patient who presented with sleep disturbance, intermittent headache, and gradual subjective cognitive decline. ^18^F-fluorodeoxyglucose (FDG) positron emission tomography (PET) showed mild FDG hypometabolism in bilateral parietal and temporal lobes. However, ^18^F-florbetaben (FBB) amyloid PET revealed diffuse amyloid retention in the lateral temporal cortex, frontal cortex, posterior cingulate cortex/precuneus, parietal cortex, and cerebellum. This finding supports the clinical significance of amyloid imaging in diagnostic work-up of early-onset Alzheimer’s disease (EOAD).

A 40-year-old female patient visited an outpatient clinic complaining of sleep disturbance, intermittent headache, and gradual subjective cognitive decline, which had started two years earlier and progressed in the past year. Working as a ballet dancer, she often repeated the mistake of forgetting dance motions, which made her quit the job and put her in a stressful situation. Her sleep disturbance and headache were not reported every time, but these symptoms occasionally appeared in stressful situations. Her mother was diagnosed with Alzheimer’s disease (AD) when she was in her 30s. There was no other past medical history. Her laboratory results, including a complete blood count, electrolytes, glucose, lipid profiles, and a thyroid function test, did not reveal any abnormalities. Her Apolipoprotein E genotyping was reported as E3/E3. Physical examination also revealed no abnormalities. A neuropsychological test battery was implemented to evaluate the patient’s cognitive status. She scored 21 on the mini-mental status examination (MMSE) and 0.5 on the clinical dementia rating scale (CDR) on the first time visit to our clinic, which suspected amnestic mild cognitive dementia. 

**Figure 1 diagnostics-13-01671-f001:**
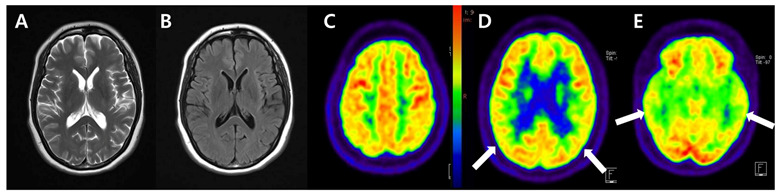
Brain MRI showed no signal abnormality or mass lesion in the whole brain parenchyma ((**A**), T2-weighted; (**B**), Fluid-attenuated inversion recovery axial image). After that, ^18^F-FDG PET/CT with intravenous injection of 226 MBq of FDG was performed to evaluate neuronal degeneration, which revealed mild glucose hypometabolism in bilateral parietotemporal lobes (white arrows in (**C**–**E**)). The attending neurologist prescribed galantamine 8 mg and decided to follow up after one year. At the next visit, she expressed the progression of memory decline and noted that recent memory loss has become more prominent.

**Figure 2 diagnostics-13-01671-f002:**
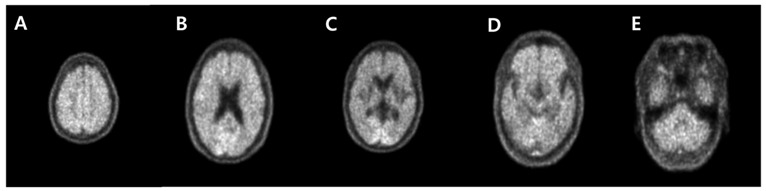
Since the early onset of dementia made the patient, an eligible candidate for an amyloid PET study, ^18^F-FBB PET/CT with intravenous injection of 300 MBq of FBB was performed. It demonstrated diffuse amyloid deposition with score three regional cortical tracer uptake (RCTU) and brain β-amyloid plaque load (BAPL) in parietal lobes (**A**), posterior cingulate cortex/precuneus (**B**), frontal lobes (**C**), lateral temporal lobes (**D**), and cerebellum (**E**) [1,2]. According to the National Institute of Neurological and Communicative Disorders and Stroke and Alzheimer’s Disease and Related Disorders Association Alzheimer’s (NINCDS-ADRDA) criteria, she was diagnosed with probable AD [3]. Based on this diagnosis, her physician prescribed donepezil 5 mg. Two years later, the neuropsychological analysis was tested again, and 20 in MMSE and 1 in CDR were recorded. The physician increased the dosage of donepezil up to 23 mg and encouraged her to engage in exercise and physical activity. Even though her cognitive decline has been tolerable, her physician recommended regular visits for her symptoms and emphasized the emotional support from her caregiver.

This report is one of the few cases of early-onset Alzheimer’s disease (EOAD) by ^18^F-FDG and ^18^F-FBB PET images (Figure 1 and Figure 2). Although this patient demonstrated the early onset of cognitive decline, it was uncertain to determine the EOAD only through ^18^F-FDG PET. However, it seemed reasonable to evaluate EOAD through ^18^F-FBB PET. Amyloid imaging has been recommended in patients with (1) early (below 65 years of age) onset of progressive dementia; (2) atypical or mixed presentation of AD; and (3) persistent or progressive unexplained mild cognitive impairment [4]. Considering her symptoms presented at an atypically early age, amyloid PET imaging was appropriately performed to investigate the etiology of her cognitive decline. It is considered that the possibility of autosomal dominant AD is more likely from her clinical history [5]. Consistent with the amyloid cascade hypothesis, it can be assumed that amyloid deposition preceded glucose hypometabolism or structural atrophy in autosomal dominant AD [6]. We focus on the additive significance of amyloid PET imaging to determine the clinical diagnosis in uncertain dementia cases even after the assessment of neurodegeneration by ^18^F-FDG PET [7,8]. Although the genotype test was lacking in this case, this report aims to help the physician determine the differential diagnosis of autosomal dominant EOAD by amyloid PET imaging. Therefore, it is important to perform amyloid PET imaging in patients with early-onset cognitive decline to optimize the management, emphasizing the implementation of clinical practice.

## Data Availability

The data that support the findings of this study are available from the corresponding author J.Y., upon reasonable request.

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
