# Peer review of "A Case Report of Early-Onset Alzheimer’s Disease Using 18F-FDG PET and 18F-FBB PET"

_diagnostics, 2023, doi:10.3390/diagnostics13101671_

Round 1

Reviewer 1 Report

The “images of interest” submitted are interesting and innovative images with enough quality and resolution and provide relevant information to be published in the Diagnostics.

Author Response

The “images of interest” submitted are interesting and innovative images with enough quality and resolution and provide relevant information to be published in the Diagnostics.

--> Thank you for kind comments. 

Reviewer 2 Report

Dear authors,

I think that the article is interesting and the case is surely of clinical interest.

I have only few suggestions:

- add the keyword "case report"

- use the same color scale (grey) for both FDG and FBB

- Please specify the administered activity (in MBq) for both FDG and FBB.

Author Response

Dear authors,

I think that the article is interesting and the case is surely of clinical interest.

I have only few suggestions:

- add the keyword "case report"

- use the same color scale (grey) for both FDG and FBB

- Please specify the administered activity (in MBq) for both FDG and FBB.

à Thank you for your comments. As you commented, we have added the keyword “case report” and the administered activity for both FDG and FBB. According to guidelines for FBB interpretation, however, FBB PET images should be displayed in the transaxial orientation using gray scale or inverse gray scale. Therefore, we illustrated in gray scale images. We also added references that corresponds to this point.  

Reviewer 3 Report

This is indeed an interesting case study. Thank you for putting the effort in to share the results. 

The only improvement I seek would be adding appropriate scale bars to Figure 1 as well as arrows pointing to the areas of hypometabolism for non-experts.

The captions are commendable as the level of detail made it almost possible to extract the entire paper's information from skimming the figures. 

Author Response

This is indeed an interesting case study. Thank you for putting the effort in to share the results. 

The only improvement I seek would be adding appropriate scale bars to Figure 1 as well as arrows pointing to the areas of hypometabolism for non-experts.

The captions are commendable as the level of detail made it almost possible to extract the entire paper's information from skimming the figures. 

--> Thank you for helpful comments. We have added scale bar in Figure 1 and pointed the areas of hypometabolism.